# Seroprevalence and risk factors of hepatitis B virus infection among healthcare workers in Africa: A systematic review and meta-analysis

**Leykun Berhanu**[1]*, **Belay Desye**[1], **Chala Daba**[1,2], **Gete Berihun**[3], **Abebe Kassa Geto**[4]

**1** Department of Environmental Health, College of Medicine and Health Sciences, Wollo University, Dessie, Ethiopia, **2** National Centre for Epidemiology and Population Health, The Australian National University, Canberra, ACT, Australia, **3** Department of Environmental Health, College of Medicine and Health Sciences, Debre Markos University, Debre Markos, Ethiopia, **4** Department of Public Health, College of Health Sciences, Woldia University, Woldia, Ethiopia

* leyberhanuwu.edu.et@gmail.com

## Abstract

### Background

Healthcare workers are at an increased risk of hepatitis B virus infection due to potential exposure to blood and other infectious materials. The infection can lead to acute liver disease and chronic liver complications such as cirrhosis and liver cancer. It can impact workforce health, leading to absenteeism, and increased healthcare costs. Hence, this study aimed to determine the seroprevalence and risk factors of the hepatitis B virus among healthcare workers in Africa.

### Materials and methods

The protocol for this systematic review and meta-analysis was registered on PROSPERO with the registration number CRD42024556654. Literatures were searched from PubMed, Science Direct, HINARI, African Online Journal, Google Scholar, Google, Semantic Scholar, and Directory of Open Access Journals using relevant search terms. The process of searching relevant articles was completed on 1 August 2024. Studies with a quality evaluation indicator score of 50% or above were included in this study. The random effect model was used to measure the pooled seroprevalence and associated factors of hepatitis B virus infection among healthcare workers in Africa. The finding of the meta-analysis was presented using forest plots with a 95% confidence interval.

### Result

Among 26 studies selected for meta-analysis, 6983 participants were included. The inclusion of 26 studies showed that the pooled prevalence of hepatitis B virus infection among healthcare workers was 17.2% (95% CI: 8.36, 26.04). Healthcare workers diagnosed with liver disease were 5.01 times more likely to having hepatitis B virus infection compared

**Data availability statement:** The data are contained within the manuscript and/or supporting information files.

**Funding:** The author(s) received no specific funding for this work.

**Competing interests:** The authors have no competing of interests.

**Abbreviations:** ELISA, Enzyme-Linked Immunosorbent Assay; HBV, Hepatitis B Virus; HCW, Healthcare Workers; HIV, Human Immune Virus; POR, Pooled Odds Ratio; WHO, World Health Organization.

to those who were not diagnosed (POR = 5.01: 95% CI; 2.25,7.77). In addition, healthcare workers who did not receive technical training were 2.70 times more likely to having HBV infection than those who received training (POR = 2.70:95% CI; 1.10, 4.30). Furthermore, healthcare workers aged 40 years and above were 2.53 times more likely to having hepatitis B virus infection than young healthcare workers (POR = 2.53: 95% CI; 1.29,3.77).

## Conclusion

The pooled prevalence of hepatitis B virus infection was high. Previously diagnosed liver diseases, the absence of technical training, and the age of healthcare workers were the factors influencing the pooled prevalence of HBV infection among healthcare workers. Hence, providing appropriate medical follow-up for healthcare workers diagnosed with liver disease, comprehensive training and education, and early detection and diagnosis of healthcare workers aged 40 years and above are the most important interventions to prevent the risk of hepatitis B virus infection.

## Introduction

Globally, hepatitis B virus is one of the most prevalent infections. The symptoms of hepatitis B virus (HBV) infection include liver inflammation, cirrhosis, and an elevated risk of liver cancer and chronic liver disease [1].

Since 1982, there has been a vaccine available worldwide to prevent this virus. The efficacy of HBV vaccine in averting infection and its long-term effects is 95% [2]. HBV infection is the main reason for liver transplants in several nations. The disease's financial toll is also significant; in the latter stages of the illness, treatments can easily run into the hundreds of thousands of dollars per patient [3,4]. After a worldwide summit in 2015, the World Health Organization (WHO) established the global initiative to combat hepatitis. The program's objectives were to treat 80% of eligible individuals with hepatitis B by 2030, reduce the number of new cases of hepatitis B by 90%, and reduce the number of hepatitis B-related deaths by 65% [5].

Unprotected sexual contact is one of the main ways that HBV is spread. The risk of contracting the virus increases when sexual activity is performed without the use of condoms, particularly if one partner is afflicted. This mode of transmission emphasizes how crucial safe sexual behavior is in lowering the risk of HBV infection. During childbirth, an infected mother's children may contract HBV. Should the mother have a high viral load at birth, this vertical transmission may happen. Pregnant women should get tested for HBV because prompt treatment, frequently in the form of vaccine and antiviral medication, can dramatically lower the risk of infection in the unborn child. HBV can potentially spread through blood products or transfusions that contain infection. Even though many nations have strict screening procedures in place to check donated blood for HBV, there is still a chance, especially in places with laxer laws. Direct contact with tainted blood or other body fluids, such as through shared needles, cuts, or open wounds, can spread HBV. This is especially important for people who use intravenous medicines or participate in activities where they could come into contact with blood. Using protective gear and maintaining good hygiene is crucial to reducing this risk [2,6–10].

According to the WHO estimates, 257 million people worldwide, or 3.5% of the total population, had a chronic HBV infection in 2015. Sixty-eight percent of the infected were from the Western Pacific and African regions. Human Immunodeficiency Virus (HIV) and HBV co-infected 2.7 million individuals. The majority of individuals who are afflicted with HBV infection were not around when the hepatitis B vaccine was first made readily accessible and

administered to infants. HBV is endemic in Africa, where an estimated 82 million people are chronically infected, with a prevalence of 6.1%. Globally, hepatitis B virus infection resulted in 1.34 million fatalities in 2015; this number was greater than that of HIV infections and on par with tuberculosis mortality [11].

According to the WHO predictions, 1.2 million new cases of chronic hepatitis B infection occur annually, impacting 254 million people worldwide in 2022. An estimated 1.1 million people died from hepatitis B virus infection in 2022, primarily from cirrhosis and hepatocellular carcinoma, or primary liver cancer [7]. Despite safe and robust HBV vaccines being available for more than 40 years, an estimated 990,000 new HBV infections and 80,000 HBV-related deaths occurred in 2019 in the WHO Africa region [12]. A systematic review and meta-analysis study done in Ghana, Cameroon, Burkina Faso, and Sudan showed that the pooled HBV prevalence was found to be 12.3% [13], 10.6%[3], 11.21%[14], and 12.07% [10], respectively.

Healthcare workers (HCWs) are particularly vulnerable to HBV infection at work because they frequently come into touch with patients' bodily fluids, including blood. According to published research, the risk of contracting HBV is four times higher among healthcare workers than it is in the general adult population who do not work in healthcare facilities [15]. Needle sticks and other sharp object injuries are the most frequent means of transmission from patients to employees, followed by mucocutaneous exposure [16]. Approximately two million healthcare workers are exposed to HBV each year, and roughly 70,000 of them contract the infection [17]. According to studies, the frequency of sharp object-related injuries among healthcare workers varies from 1.4 to 9.5 per 100 workers annually, leading to 0.42 HBV infections per 100 sharp object injuries annually [2].

WHO has advised vaccination for all newborns, all children and teenagers under the age of 18, and all high-risk groups, which include those who engage in high-risk sexual behavior, those who inject drugs, those who frequently need blood or blood products, those who receive solid organ transplants, those who are at occupational risk of contracting the hepatitis B virus, and visitors to nations with high rates of hepatitis B infection [2].

While existing systematic reviews and meta-analyses provide a pooled prevalence, they often overlook specific associated factors impacting HBV prevalence among HCWs. The African continent is diverse, and factors influencing HBV prevalence may vary significantly between regions. This variation needs to be explored in detail. This systematic review and meta-analysis study synthesizes data from multiple studies, providing a more robust estimate of seroprevalence than any single study could offer. By aggregating data, this study identifies trends and patterns across different populations and settings that may not be evident in smaller primary studies. The analysis explores subgroups that may be underrepresented in individual studies. The findings can inform evidence-based guidelines and policies for preventing HBV among healthcare workers in Africa.

Understanding the pooled HBV prevalence among HCWs in Africa contributes to global health knowledge, particularly regarding infectious diseases that affect health systems worldwide. Findings can inform international health organizations and governments about the need for targeted interventions and vaccination strategies, potentially reducing HBV transmission globally. Data from this study can guide resource allocation for HBV prevention and treatment programs, ensuring that areas with higher prevalence receive adequate support.

Identifying risk factors helps develop strategies to protect HCWs, who are vital for the health of communities, thereby enhancing healthcare delivery. By understanding the transmission dynamics of HBV among HCWs, interventions can be designed to reduce community transmission, benefiting public health overall. Insights from the study can lead to culturally tailored interventions that resonate with local practices and beliefs, improving acceptance and effectiveness.

The study identifies knowledge gaps, guiding future research efforts and encouraging studies that address specific factors influencing HBV prevalence. It provides a methodological template for researchers interested in conducting similar studies on other infectious diseases or health issues. The finding can serve as a strong evidence base for funding applications aimed at HBV research and intervention programs. Therefore, this systematic review and meta-analysis study aimed to determine the pooled prevalence and risk factors of HBV among HCWs in Africa.

## Materials and methods

### Protocol registration

The protocol for this systematic review and meta-analysis was registered on Prospero on 21/06/2024 with the registration number CRD42024556654.

### Search strategy for article identification

A comprehensive systematic literature search was conducted in electronic databases including PubMed, Science Direct, HINARI, African Online Journal, Google Scholar, Google, Semantic Scholar, and Directory of Open Access Journal, using relevant search terms. Moreover, relevant articles were collected through literature from the retrieved articles. The process of searching relevant articles was completed on 1 August 2024. The initial search was conducted on advanced PubMed databases using Boolean operators like "OR" and "AND". To improve the review's repeatability, the date of searching for papers in each database was noted at the time of searching. The relevant literature was searched on PubMed using the following search terms, "(Hepatitis B" OR "Hepatitis B virus" OR "Hepatitis B virus infection" OR "Serum hepatitis" OR "HBV infection" OR "Hepatitis B surface antigen (HBsAg)" OR "Chronic hepatitis B" OR "Acute hepatitis B" OR "Viral hepatitis B" OR "Hepatitis B virus (HBV) infection)" AND "(Contributing factors" OR "Influencing factors" OR "Detrimental factors" OR "Associated factors" OR "Risk determinants)" AND "(Healthcare workers" OR "Healthcare personnel" OR "Health service workers" OR "Health care team" OR "Patient care staff)" AND (Algeria OR Angola OR Benin OR Botswana OR "Burkina Faso" OR Burundi OR "Cabo Verde" OR Cameroon OR "Central African Republic" OR Chad OR Comoros OR Congo OR "Democratic Republic of the Congo" OR "Republic of the Cote d'Ivoire" OR Djibouti OR Egypt OR "Equatorial Guinea" OR Eritrea OR Eswatini OR Ethiopia OR Gabon OR Gambia OR Ghana OR Guinea OR "Guinea-Bissau" OR Kenya OR Lesotho OR Liberia OR Libya OR Madagascar OR Malawi OR Mali OR Mauritania OR Mauritius OR Morocco OR Mozambique OR Namibia OR Niger OR Nigeria OR Rwanda OR "Sao Tome and Principe" OR Senegal OR Seychelles OR "Sierra Leone" OR Somalia OR "South Africa" OR "South Sudan" OR Sudan OR Tanzania OR Togo OR Tunisia OR Uganda OR Zambia OR Zimbabwe). The searches were restricted using language, article type, text availability and species. Publication date was not used to restrict searching. In addition, by incorporating key terms searches on additional databases were also conducted (S1 Table).

### Eligibility criteria

In this systematic review and meta-analysis study, all articles whose study design is observational i, e, cross-sectional, cohort, and case-control, articles done to determine the prevalence of HBV and or risk factors among all healthcare workers were included. In addition, all articles written in English were included in this systematic review and meta-analysis.

However, those reports such as abstracts, books, letters, reports, reviews, and guidelines were excluded. Moreover, studies conducted out of the African continent and those written in a language other than English were also excluded. In addition, animal studies, and studies done other than English language were excluded from this systematic review and meta-analysis study.

## Outcome assessment

The primary goal of this study was to determine the pooled seroprevalence of HBV infection and associated factors among HCWs in Africa. The number of research participants with HBV infection divided by the actual sample size was used to estimate the prevalence which was then multiplied by 100. Furthermore, a pooled odds ratio with 95% CI was calculated to determine the factors associated with the prevalence of HBV. Consequently, a systematic review and meta-analysis were done.

## Operational definitions

**Presence of HBV infection** Is the presence of serum HBsAg (current infection) and/or anti HBC (current or past resolved infection) considered evidence for exposure to HBV (recent infection or chronic carrier) [18,19].

**HCWs** are all people engaged in work actions whose primary intent is to improve health, including doctors, nurses, midwives, public health professionals, laboratory technicians, health technicians, medical and non-medical technicians, personal care workers, community health workers, healers and traditional medicine practitioners, cleaners, drivers, hospital administrators, district health managers and social workers, and other occupational groups in health related activities as defined by the International Standard Classification of Occupations [20].

## Study selection process

Two authors namely LB and AKG independently screened all articles based on title, abstracts, and full texts. The screened articles were compiled by LB and AKG to get them ready for additional screening and selection processes. When there was disagreement, a third reviewer, GB, was consulted in order to reach a consensus. The study selection process has been simplified using the 2020 PRISMA flow diagram (Fig 1).

## Data extraction process and data items

All of the papers were exported to Endnote version 20 software once pertinent literature was scanned. The amount of duplicate articles deleted from each database was tracked. After deleting all duplicates, the studies were assessed based on titles and abstracts against predetermined study inclusion criteria. A data extraction template was created, which included the author's name, year of publication, sample size, country, geographic location, type of study design, response rate, seroprevalence of HBV, potential risk factors for HBV, adjusted odds ratio, and confidence interval for each of the predictors were noted by using a standardized data collection form.

In addition, sex, blood and blood product contact, needle stick injury, sharp injury, diagnosed liver disease, blood transfusion, splash of blood and body fluids, tattoo, utilization of PPE, technical training, age, type of staff, duration of employment, mucocutaneous, hospitalization, dental procedure, unprotected sex, multiple sex partner, contact history, marital status, staff group, circumcised, year of service, ever vaccinated, religion were extracted from the included studies.

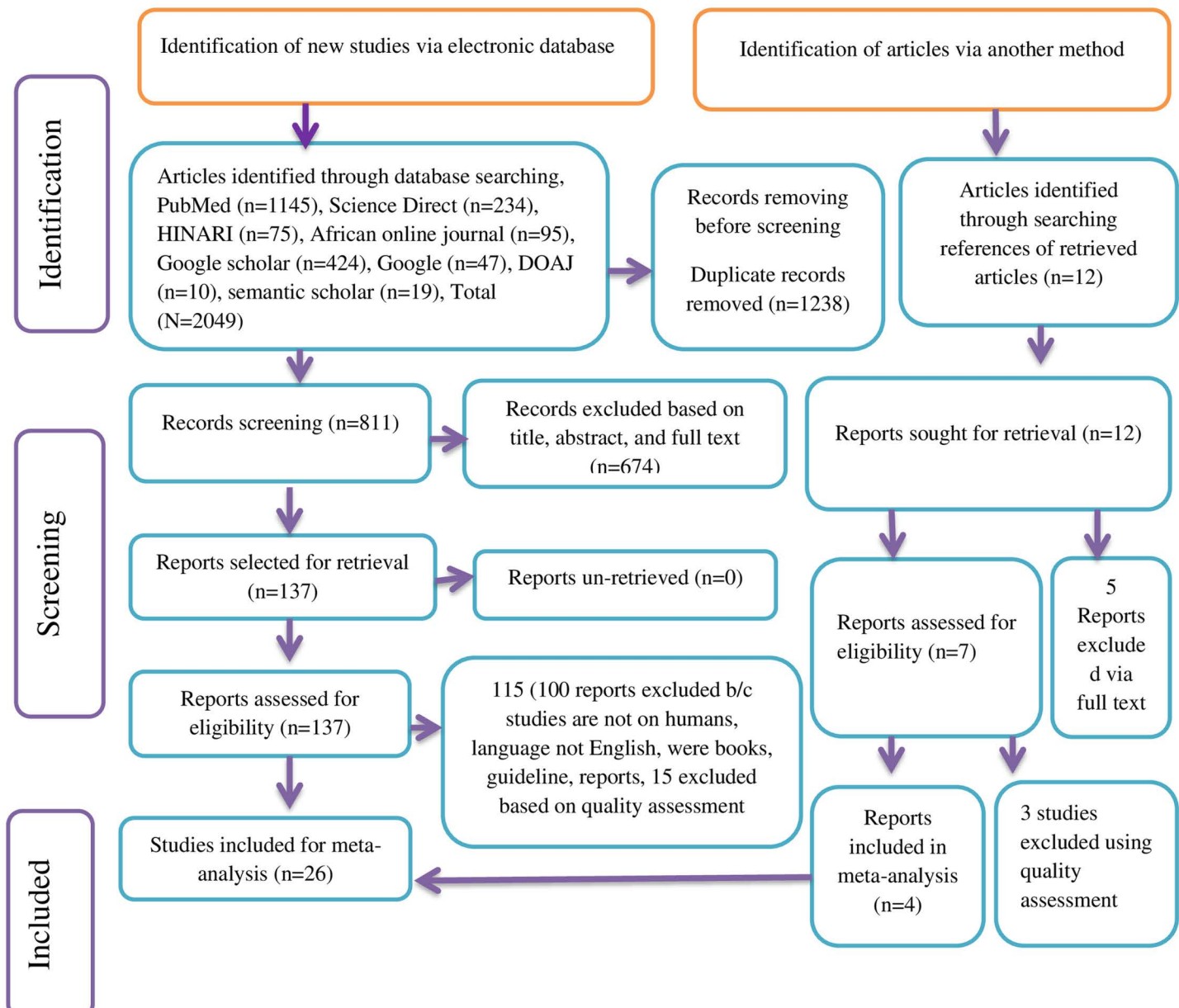

**Fig 1. PRISMA flow diagram showing the selection of studies for systematic review and meta-analysis on the seroprevalence and risk factors of HBV among HCWs in Africa, 2024.**

## Study quality and bias assessment

The Joanna Briggs Institute (JBI) quality rating instrument was used to assess the quality of papers for consideration in the systematic review and meta-analysis. LB, AKG, and GB independently performed study quality assessments using JBI criteria for a cross-sectional study. The criteria measured each of the articles out of 100 percent. After a thorough review, studies with a score of 50% and above for each study were identified as having low risk of quality. Any disagreement in the quality assessment process was resolved by taking the mean score of the quality assessors. Studies with a quality evaluation indicator score of 50% or above were

included in the study [21] (S2 Table). To assess publication bias impartially, Egger's and Begg's tests were applied. A funnel plot was also created to subjectively assess the presence of publication bias in articles included in this review.

## Data analysis and presentation

The data was extracted using an Excel sheet and exported to STATA version 17 software for analysis. The presence and level of heterogeneity among the selected studies were assessed using $I^2$ and p-values. The heterogeneity of the studies was considered high risk ($I^2 = 99.7\%$, $p < 0.0001$). To identify the source of heterogeneity, subgroup analysis was performed based on geographic location, type of healthcare facility, sample size category, and HBV detection method. Two categories were used to classify the sample size: less than 269 and higher than or equal to 269. The average sample size of the included studies serves as the basis for the categorization threshold. Due to the presence of high heterogeneity among the included studies, a random effect model was used to measure the pooled seroprevalence and associated factors of HBV infection among HCWs in Africa. To identify the significant factors associated with heterogeneity, a univariate meta-regression analysis was performed. A trim and fill study was done to identify the cause of this bias. Sensitivity analysis was performed to determine a single study effect on the pooled seroprevalence of HBV. The finding of the meta-analysis was presented using a table, graph, and forest plot with a 95% confidence interval. A p-value of less than 0.05 was considered statistically significant.

## Results

### Characteristics of the included studies

In this study, a total of 6983 participants were included in the study. All the included studies are cross-sectional. Among 26 studies included in the review, 6983 HCWs were screened for HBV infection. Regarding the country of publication, 7 studies were done in Ethiopia [22–28], five in Nigeria [18,29–32], two studies each from Uganda [33,34], Kenya [35,36], Tanzania [37,38], Sierra Leone [39,40], and one study each from Cameron [41], Ghana [42], Rwanda [43], Congo DR [19], Mozambique [44], and Sudan [45]. The highest seroprevalence was reported in Niger with the rate of 86.7% [31] while the lowest was reported in Nigeria with the rate of 1.1% [29]. Regarding the HBV detection method, most studies 17(65.4%) used Enzyme-Linked Immunosorbent Assay (ELIA) to detect the presence of current infection or life time exposure with HBV (Table 1).

## Meta-analysis

### Pooled prevalence

Due to the high heterogeneity among the included studies, a random effect model was used to estimate the pooled prevalence of HBV among healthcare workers in Africa. Accordingly, the inclusion of twenty-six studies confirmed that the pooled prevalence of HBV among healthcare workers in Africa was 17.2% (95% CI: 8.36, 26.04). There was a high heterogeneity level among the included studies ($I^2 = 99.7\%$, $p < 0.0001$) (Fig 2).

### Publication bias assessment

The presence of possible publication bias was assessed using visual observation from a funnel plot and statistical method using Egger's test. From the funnel plot, it is observed that the studies are not distributed symmetrically around the mean effect size, with smaller studies

**Table 1. Characteristics of the studies included studying the prevalence of HBV and associated factors among HCWs in Africa, 2024.**

| Author Year | Country | Geographic location | Sample size | Dis-eased | Prevalence of HBV (%) | 95% CI | Detection method | JBI score (100%) |
|---|---|---|---|---|---|---|---|---|
| Akalu et al, 2016[22] | Ethiopia | East Africa | 313 | 8 | 2.6 | 0.84,4.36 | CMIA | 75 |
| Akazong et al, 2020[41] | Cameron | Central Africa | 398 | 42 | 10.6 | 7.58, 13.62 | ELISA | 50 |
| Alese et al,2016[29] | Nigeria | West Africa | 187 | 2 | 1.1 | 0, 2.60 | ELISA | 62.5 |
| Amsalu et al, 2016[23] | Ethiopia | East Africa | 152 | 62 | 40.7 | 32.88, 48.52 | RIT | 62.5 |
| Ayele et al, 2023[24] | Ethiopia | East Africa | 276 | 14 | 5.1 | 2.49, 7.71 | ELISA | 75 |
| Belo, 2000[30] | Nigeria | West Africa | 167 | 43 | 25.7 | 19.06, 32.34 | ELISA | 62.5 |
| Braka et al, 2006[33] | Uganda | East Africa | 311 | 187 | 60.1 | 54.65, 65.55 | ELISA | 50 |
| Efua et al, 2023[42] | Ghana | West Africa | 363 | 21 | 5.8 | 3.40, 8.20 | ELISA | 75 |
| Elikwu et al, 2016[18] | Nigeria | West Africa | 100 | 7 | 7 | 2.0%, 12.0 | ELISA | 50 |
| Yilma et al, 2021[25] | Ethiopia | East Africa | 457 | 8 | 1.8 | 0.58, 3.02 | ELISA | 62.5 |
| Kateera et al, 2015[43] | Rwanda | East Africa | 378 | 11 | 2.9 | 1.21, 4.59 | RIT | 50 |
| Kisangau et al, 2019[35] | Kenya | East Africa | 295 | 13 | 4 | 1.76, 6.24 | ELISA | 62.5 |
| Lungosi et al, 2019[19] | Congo DR | East Africa | 97 | 18 | 18.6 | 10.86, 26.34 | ELISA | 50 |
| Mabunda et al, 2022[44]) | Mozambique | East Africa | 315 | 16 | 5.1 | 2.67, 7.53 | ELISA | 62.5 |
| Machang et al, 2017[37] | Tanzania | East Africa | 76 | 9 | 11.9 | 4.65, 19.15 | ELISA | 50 |
| Massaquoi et al, 2018[39] | Sierra Leone | West Africa | 447 | 39 | 8.7 | 6.09, 11.31 | EIA | 50 |
| Mbaawuag et al, 2019[31] | Niger | West Africa | 225 | 221 | 86.7 | 82.25, 91.15 | ELISA | 62.5 |
| Mboya et al, 2023[36] | Kenya | East Africa | 192 | 36 | 18.75 | 13.21, 24.29 | EIA | 62.5 |
| Mekonnen et al, 2015[26] | Ethiopia | East Africa | 252 | 9 | 3.57 | 1.28, 5.86 | ELISA | 62.5 |
| Mengiste et al, 2021[27] | Ethiopia | East Africa | 260 | 53 | 20.4 | 15.50, 25.30 | RIT | 62.5 |
| Nail et al, 2008[45] | Sudan | East Africa | 211 | 5 | 2.4 | 1.28, 5.86 | ELISA | 50 |
| Qin et al, 2018[40] | Sierra Leone | West Africa | 211 | 21 | 10 | 5.95, 14.05 | RIT | 50 |
| Sani et al, 2018[32] | Nigeria | West Africa | 100 | 19 | 19 | 11.33, 26.67 | ELISA | 50 |
| Shao et al, 2018[38] | Tanzania | East Africa | 442 | 25 | 5.7 | 3.54, 7.86 | Laborex HBsAg rapid test | 62.5 |
| Yizengaw et al, 2018[28] | Ethiopia | East Africa | 388 | 10 | 2.6 | 1.02, 4.18 | ELISA | 75 |
| Ziraba et al, 2010[34] | Uganda | East Africa | 370 | 208 | 56.2 | 51.14, 61.26 | ELISA | 75 |

spread more widely at the bottom and larger studies clustered more closely at the top; this suggests the presence of publication bias (Fig 3).

According to the Egger's statistical test publication bias was observed (p < 0.001) (Table 2).

The finding from trim and fill analysis indicates notable variation in the newly estimated pooled odds ratio, denoted as the adjusted point estimate (OR = 1.205; 95% CI: 0.712, 1.698) when compared to the initial or observed point estimate (OR = 1.759); 95% CI: 1.307, 2.211 (Fig 4).

## Sensitivity analysis

Sensitivity analysis was performed to determine the effect of a single study on the pooled prevalence of HBV among the included studies. The finding showed that the pooled prevalence of HBV is not influenced by a single study (p < 0.0001) (Fig 5).

## Subgroup analysis

Subgroup analysis was performed using geographic location, type of healthcare facility, sample size category, and HBV detection method.

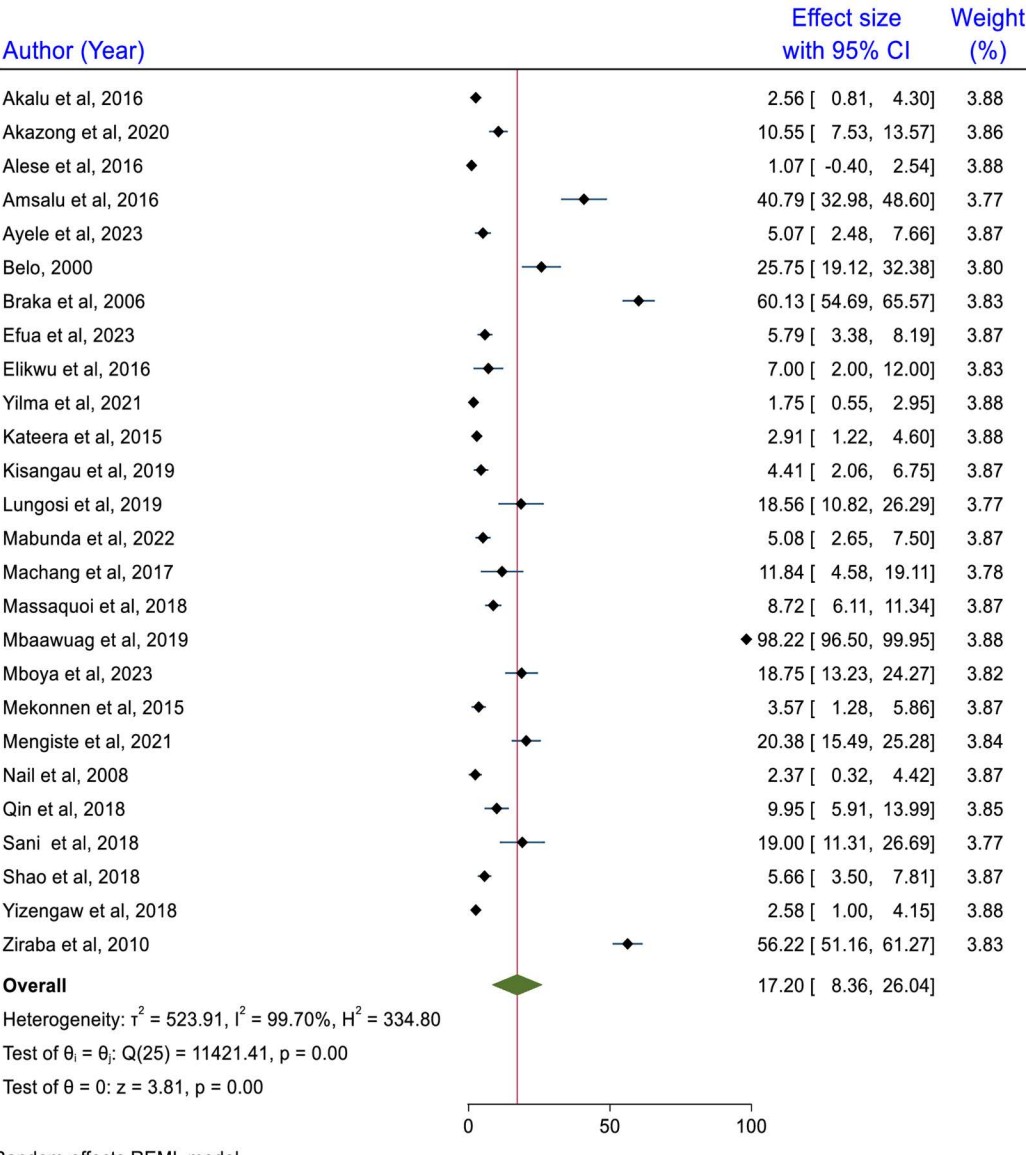

**Fig 2. Pooled seroprevalence of HBV among HCWs in Africa, 2024.**

## Subgroup analysis based on geographic location

The subgroup analysis based on the geographic location showed that the highest pooled prevalence of HBV was reported among studies done in West Africa with a prevalence of 21.96% (95% CI: -0.17, 44.08), the lowest was reported in Central Africa where the pooled prevalence of HBV infection was reported to be 10.55% (95% CI: 7.73, 13.57). However, the pooled prevalence of HBV did not show statistically significant differences among the categories (p = 0.39) (Fig 6).

## Subgroup analysis based on sample size category

As shown below in the figure, the highest pooled prevalence of HBV among studies with a sample size of less than 269 was 21.33% (95% CI: 7.31, 35.35) while the lowest prevalence of

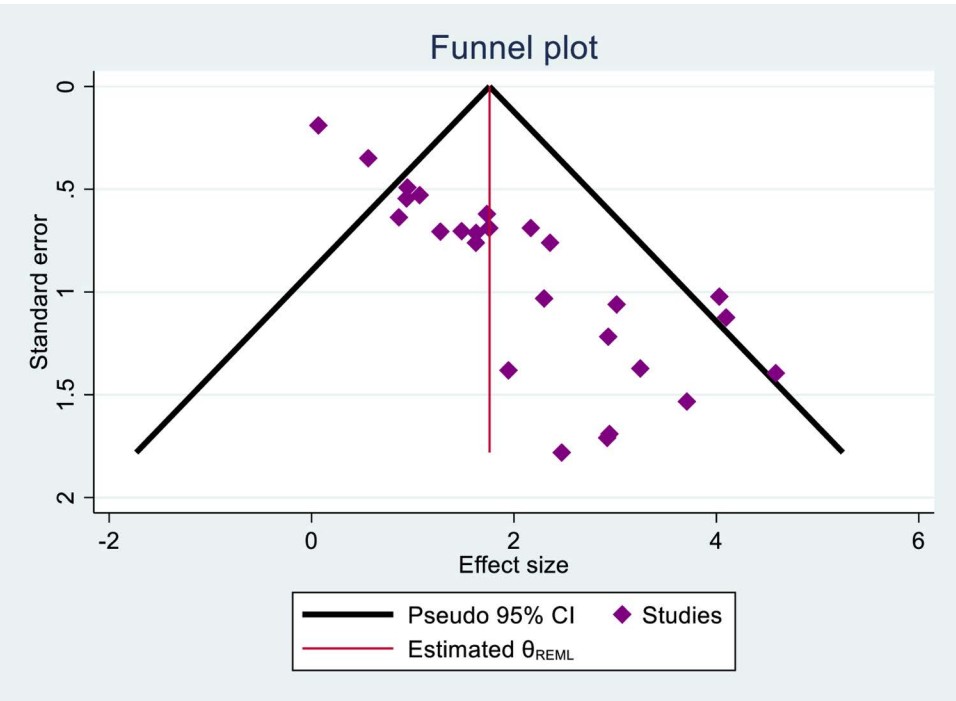

**Fig 3. Funnel plots showing the presence of publication bias among the included studies, 2024.**

**Table 2. Egger's statistical test result showing the presence of publication bias among the included studies, 2024.**

| Study effect | Coefficient | Standard error | P-value | 95% CI |
|---|---|---|---|---|
| Slope | -0.4067322 | 0.1458492 | 0.010 | -0.7077502, -0.1057141 |
| Bias | 2.804945 | 0.2445805 | <0.001* | 2.300155, 3.309734 |

*Test of HO: no small study effect, p = 0.000,*

*\* Indicates the presence of statistical significance*

HBV was found to be 13.08% (95% CI: 2.25, 23.92). However, the difference is not statistically significant (p = 0.36) (Fig 7).

## Subgroup analysis based on HBV detection method

The subgroup analysis based on the detection method revealed that the highest pooled HBV prevalence was reported among studies using ELISA as an HBV detection method with a prevalence of 19.57% (95% CI: 6.69, 32.45), and the lowest was reported among studies who used CMIA with the prevalence of 2.56% (95% CI: 0.81, 4.30). The detection ability of the test method employed among the included studies was CMIA, Laboarex HBsAg, RIT, and ELISA, indicating the high detection ability of ELISA. The difference in the pooled HBV prevalence among the different detection methods is significant (p < 0.001) (Fig 8).

## Subgroup analysis based on type of health facility

We looked at the pooled prevalence of HBV in public, private, and combined healthcare settings. The finding indicated that the highest pooled prevalence of HBV was reported among public health facilites where the value is reported to be 18.61% (95% CI: 8.76, 28.47). The

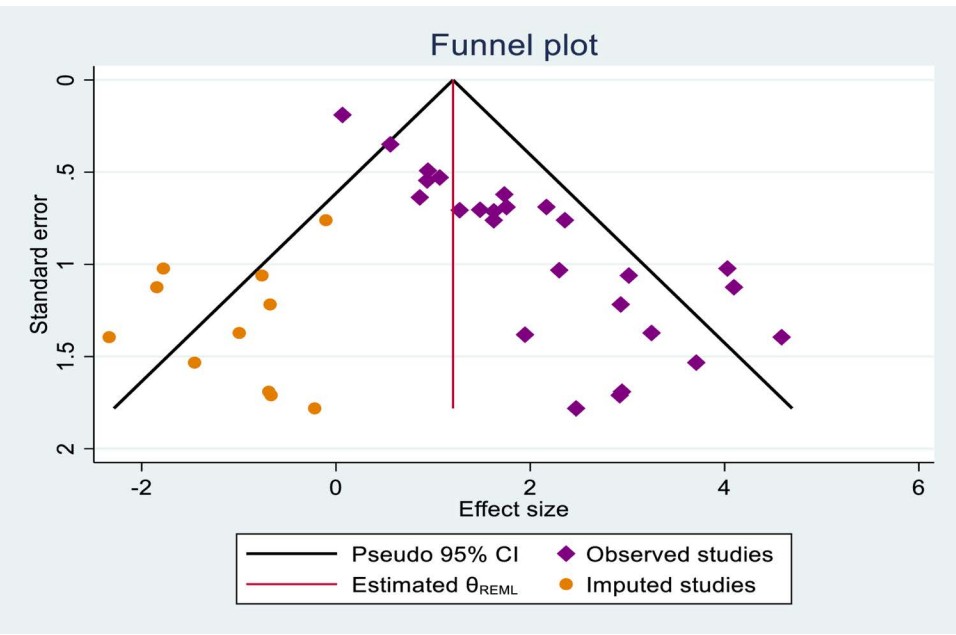

**Fig 4. The funnel plot of a simulated meta-analysis after including eleven hypothetical studies to control the publication bias observed in this systematic review and meta-analysis study, 2024.**

lowest pooled HBV was reported among studies focused on private health facilitates with the value found to be 7.0% (95% CI: 2.00, 12.00). However, the mean difference did not show a significant difference (p = 0.10) (Fig 9).

## Meta-regression

To identify the potential sources of heterogeneity, meta-regression analysis was performed by considering the geographic location, type of healthcare facility, sample size category, and HBV detection method. The finding revealed that the geographic location, type of healthcare facility, sample size category, and HBV detection method did not significantly contribute to heterogeneity among the included studies (Table 3).

## Factors associated with HBV among HCWs

This systematic review and meta-analysis study revealed that, diagnosed liver disease, age of health care workers, and the presence of technical training were the factors significantly associated with the pooled prevalence of HBV among HCWs in Africa. The association between diagnosed liver disease and the pooled prevalence of HBV infection was studied using five studies [22,23,25,28,43]. The results showed that the risk of contracting HBV infection was 5.01 times higher for healthcare workers with liver disease diagnoses than for those without (POR = 5.01:95% CI: 2.25,7.77). In addition, the association between the age of HCWs and the pooled prevalence of HBV infection was studied by using three studies [19,23,34]. The finding showed that those HCWs who are greater than or equal to 40 years of age were 2.53 times more likely to having HBV infection as compared to those who are between the ages of 18 -25 (POR = 2.53, 95% CI: 1.29, 3.77). Moreover, the association between technical training and the pooled prevalence of HBV was studied by including four studies [24,25,34,42]. The finding showed that HCWs who did not have technical training about HBV were 1.02 times more

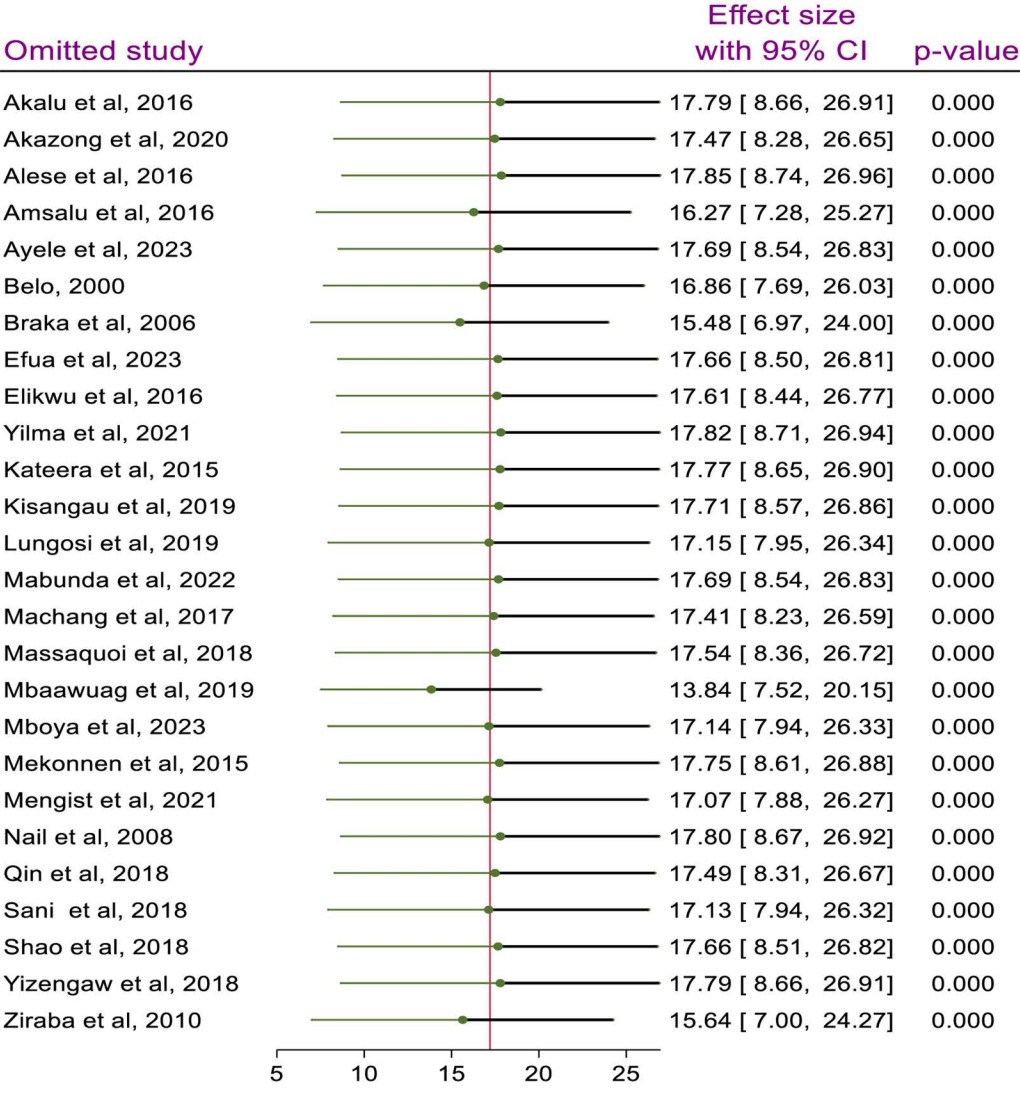

**Fig 5. Sensitivity analysis result of the included studies, 2024.**

likely to having HBV infection as compared to those who took the training (POR = 1.02: 95% CI; 1.02, 1.03) (Table 4).

## Discussion

HBV is one of the key etiological agents for liver diseases, including chronic hepatitis, liver cirrhosis, and liver cancer. It is the second most common human carcinogen after tobacco [46,47]. As a result, the WHO established a plan to eradicate hepatitis as a public health hazard by 2030, and the UN Sustainable Development Goals recommended "combating hepatitis" as an SDG-2030 target in 2015 [48].

This systematic review and meta-analysis was done to determine the pooled prevalence of HBV among HCWs in Africa. The finding showed that the pooled prevalence of HBV among HCWs was 17.2% (95% CI: 8.36, 26.04). The present reported HBV prevalence is higher as

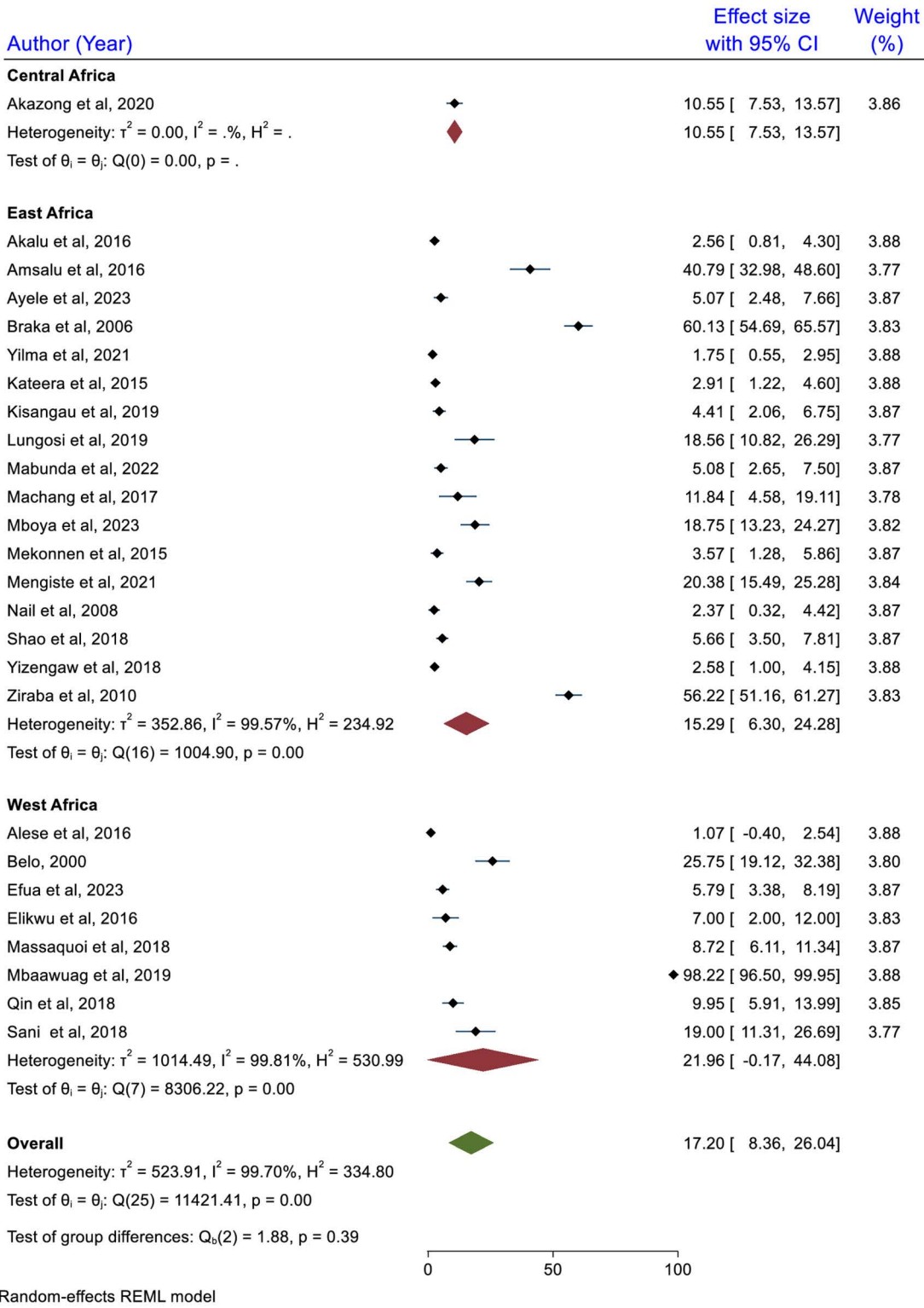

**Fig 6. Subgroup analysis result based on geographic location, 2024.**

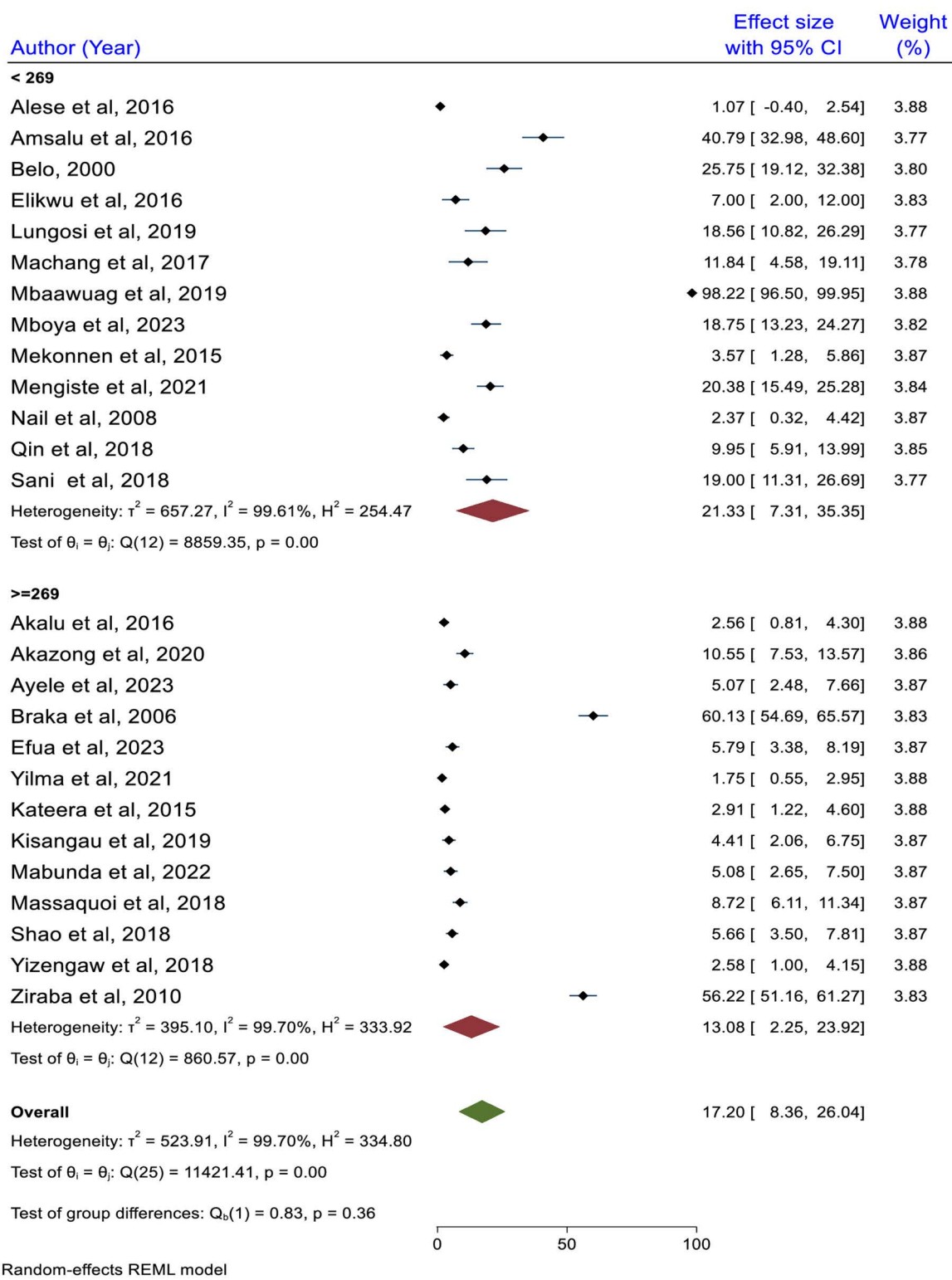

Fig 7.  Subgroup analysis result based on sample size category, 2024.

**Fig 8. Subgroup analysis result based on HBV detection method, 2024.**

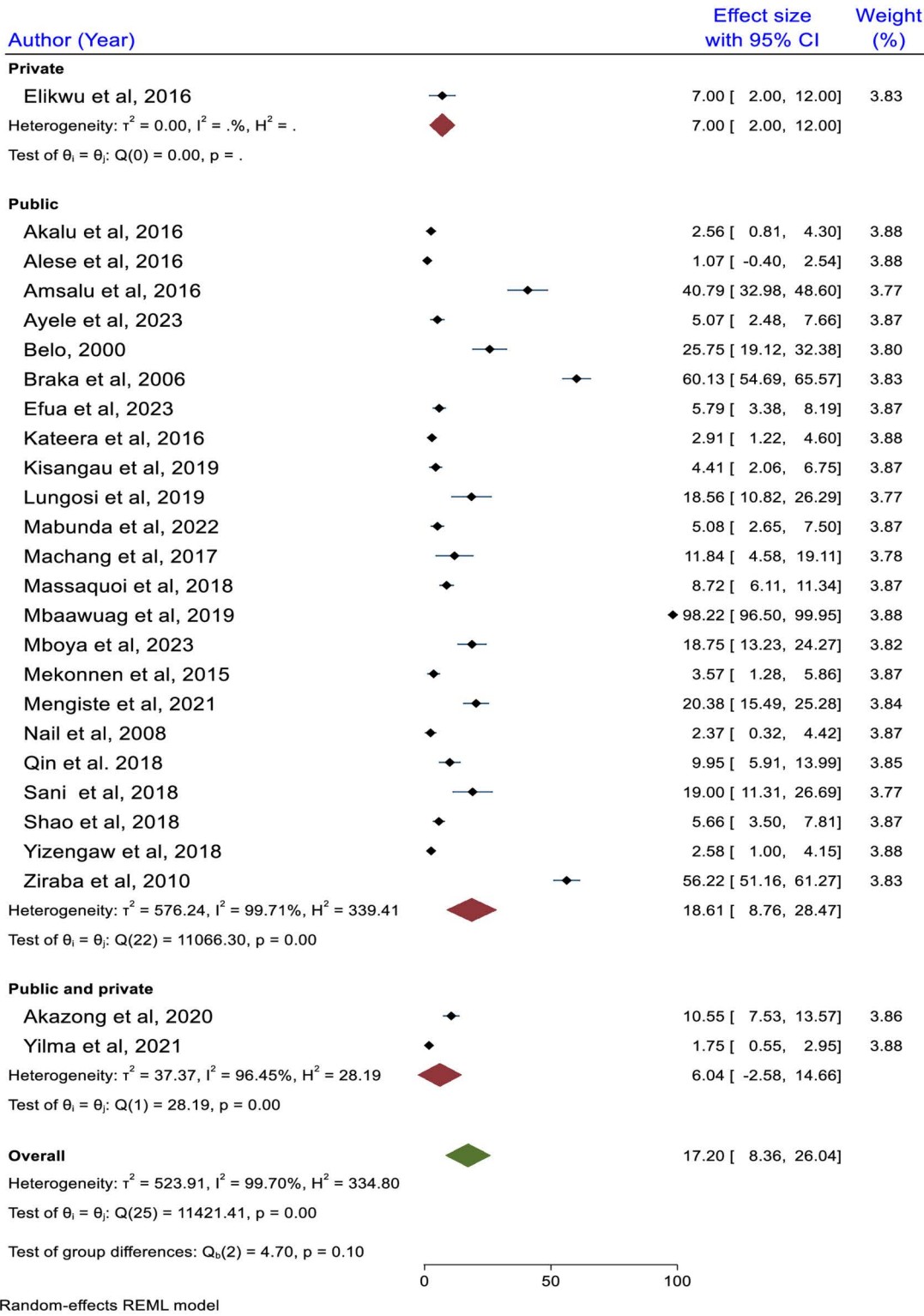

**Fig 9. Subgroup analysis result based on type of health facility, 2024.**

**Table 3. A univariate meta-regression analyses to identify the factors associated with the heterogeneity of studies included to determine the pooled prevalence of HBV among HCWs in Africa, 2024.**

| Variables | Coefficient | Standard error | P-value | 95% CI |
|---|---|---|---|---|
| Geographic location | 3.631895 | 8.501657 | 0.669 | -13.03105,20.29484 |
| Type of facility | 5.741674 | 4.821552 | 0.234 | -3.708395,15.19174 |
| Sample size category | 9.300629 | 9.49476 | 0.327 | -9.308759,27.91002 |
| Detection method | -0.64021 | 5.655027 | 0.910 | -11.72386,10.44344 |

**Table 4. Risk factors associated with the pooled prevalence of HBV infection among HCWs in Africa, 2024.**

| List of variables | Number of participant | No of studies | Pooled Odds ratio (95% CI) | Heterogeneity | |
|---|---|---|---|---|---|
| | | | | I² | P-value |
| Sex (Female)[19,22,27,34,36] | 1232 | 5 | 1.02 (0.55,1.50) | 0.0 | 0.416 |
| Blood and blood product contact (Yes)[22,24,27] | 849 | 3 | 1.71 (0.24,3.18) | 0.0 | 0.683 |
| Needle stick injury (Yes)[22,24,25,28,42] | 1797 | 5 | 1.74 (-0.13, 3.61) | 0.0 | 0.804 |
| Sharp injury (Yes)[22,23,27,40] | 936 | 4 | 0.84 (0.48, 1.20) | 27.3 | 0.248 |
| Diagnosed liver disease (Yes)[22,23,25,28,43] | 1688 | 5 | 5.01(2.25,7.77)* | 40.9 | 0.149 |
| Blood transfusion (Yes)[24,27,28,38,43] | 1744 | 5 | 1.80 (-0.23,3.84) | 0.0 | 0.786 |
| Splash of blood and body fluids(Yes) [22,25] | 770 | 2 | 1.21 (0.56,1.86) | 0.0 | 0.476 |
| Tattoo (Yes) [22–24,27] | 1001 | 4 | 1.85 (0.65,3.05) | 0.0 | 0.394 |
| Utilization of PPE (No) [24,27] | 536 | 2 | 1.40 (-0.32,3.12) | 0.0 | 0.514 |
| Technical training (No)[24,34,42] | 1009 | 3 | 2.70 (1.10,4.30)* | 0.0 | 0.942 |
| Age (30-39 years)[34,36,43] | 940 | 3 | 1.15 (0.25,2.05) | 0.0 | 0.619 |
| Age (>=40 years) [19,23,34] | 619 | 3 | 2.53 (1.29,3.77)* | 0.0 | 0.932 |
| Age (40-49)[36,43] | 570 | 2 | 1.59 (-0.37,3.55) | 0.0 | 0.722 |
| Type of staff (Others)[34,36,42] | 925 | 3 | 0.54 (-0.15,1.23) | 19.5 | 0.289 |
| Duration of employment (>=10 years)[34,42] | 733 | 2 | 2.24 (-0.28,4.76) | 0.0 | 0.776 |
| Mucocutaneous (not exposed) (23, 34, 42) | 885 | 3 | 1.03 (0.38,1.68) | 0.0 | 0.498 |
| Hospitalization (Yes)[25,28] | 845 | 2 | 1.83 (-1.01,4.67) | 0.0 | 0.734 |
| Dental procedure (Yes)[23,25,28,43] | 1375 | 4 | 0.59 (0.07,1.12) | 33.2 | 0.213 |
| Unprotected sex (Yes) [25,28] | 845 | 2 | 2.83 (-1.76,7.43) | 0.0 | 0.657 |
| Multiple sex partner (Yes)[19,23,25,28] | 1094 | 4 | 2.33 (0.45,4.22) | 43.1 | 0.153 |
| Contact history (Yes)[25,28] | 845 | 2 | 8.11 (-4.49,20.71) | 0.0 | 0.585 |
| Married (Yes)[23,34,36,43] | 1092 | 4 | 0.58 (0.11,1.06) | 37.9 | 0.185 |
| Staff group (Clinical) [34,43] | 748 | 2 | 1.08 (-0.51,2.67) | 0.0 | 0.658 |
| Staff group (cleaners)[36,43] | 567 | 2 | 0.83 (-1.09,2.75) | 0.0 | 0.996 |
| Circumcised (Male)[28,43] | 766 | 2 | 1.41 (-1.10,3.93) | 0.0 | 0.741 |
| Profession (Nurse)[19,34,36] | 659 | 3 | 0.83 (0.59,1.08) | 76.8 | 0.013 |
| Profession (Physician) [19,40] | 308 | 2 | 2.37 (-0.98,5.73) | 0.0 | 0.846 |
| Profession (Laboratory technician)[19,34,36] | 659 | 3 | 1.33 (0.27,2.39) | 69.3 | 0.039 |
| Year of service (>10 years)[23,36] | 344 | 2 | 2.13 (-8.91,13.18) | 0.0 | 0.878 |
| Ever vaccinated (Yes) [27,28,34,38,40] | 1671 | 5 | 1.78 (-0.40,3.96) | 0.0 | 0.428 |
| Religion (Christian) [34,38] | 812 | 2 | 1.10 (0.18,2.3) | 0.0 | 0.951 |

*indicates the presence of statistically significant association

compared to the global systematic review and meta-analysis with a pooled prevalence estimate of 3.61 reported a lower pooled prevalence [49]. Another systematic review and meta-analysis studies reported a lower prevalence of HBV. For example, in Ghana the prevalence of HBV was (12.3%) [13] in Cameroon (10.6%) [3], Burkina Faso (11.21%)[14], Sudan (12.07%)[10], and in Nigeria (13.6% and 8.4%) [50,51]. There are several risk factors associated with high seroprevalence of HBV infection in a healthcare setting, including accidental exposure to blood and blood products, occupational injuries from sharp objects, such as needle sticks, lack of expertise or practice with HBV infection, and no vaccination history [52–56]. In addition, the lack of inexpensive diagnostics, the lack of domestic funding for hepatitis care, the complexity of treatment protocols developed by high-income countries, and the inadequate state of the healthcare system are major factors for the increased HBV infection in the African context [57]. The studies reporting HBsAg and Anti-HBC that we have included in our systematic review and meta-analysis include healthcare workers who were infected or who had recovered from infection. Compared to those investigations and reports, this will raise the pooled prevalence of HBV infection among HCW [58].

In Europe, a very low prevalence of 0.6% HBV infection was reported [59]. This might be the extensive HBV vaccination campaigns throughout Europe, which have lowered the disease's incidence. The HBV vaccine is now part of the standard pediatric immunization regimen in many European nations. Europe enjoys a higher socioeconomic standing, better access to healthcare, and better sanitation and cleanliness standards. These elements can lessen the spread of HBV, which is mostly contracted by coming into contact with contaminated bodily fluids or blood. In Europe, intravenous drug use and sexual contact are the main ways that HBV is spread, and these can be more successfully addressed by public health initiatives. More extensive HBV screening systems have been put in place in Europe, and they can detect and treat infected people while lowering the chance of future transmission. In general, liver transplant services and antiviral drugs are more easily accessible in Europe for HBV treatment [60,61].

The present finding is also higher than that of a systematic review and meta-analysis of epidemiological studies prevalence done in East Africa 6.025% [62], East Asia 9.5% [63] and developing countries 3.6% [63]. The possible reason for the variation may be attributed to the difference in target population. Similarly systematic literature review and meta-analysis of 3740 studies and 231 million people in China reported that the prevalence of HBV was 9.6%, indicating a significant reduction in HBV infection [48]. With HBV vaccinations being a part of the national immunization program since the 1980s, China has attained a high vaccination rate. Compared to many African countries, China offers a wider range of healthcare services, including HBV diagnosis, treatment, and screening. A greater ability to obtain healthcare and preventative measures can be made possible by China's notable economic growth and improvements in living conditions [64].

However, a higher prevalence of HBV infection was reported in a systematic review and meta-analysis study in Somalia with the prevalence of HBV reported to be 19.0% [65]. This might be because of the brittle healthcare system in Somalia, which has existed for decades due to political unrest and conflict. Particularly in rural areas, access to HBV immunization, screening, and treatment is restricted [66].

The subgroup analysis based on the detection method revealed that the highest pooled HBV prevalence was reported among studies using ELISA as an HBV detection method with a prevalence of 19.57% (95% CI: 6.69, 32.45), and the lowest was reported among studies who used CMIA with the prevalence of 2.56% (95% CI: 0.81, 4.30). The difference is significant ($p < 0.001$). This may be because ELISA tests, which have an extremely sensitive and specific design, are able to detect low concentrations of viral antigens or antibodies in the serum. This screening method helps identify infections that are early or low-level [67].

This study revealed that health care workers who are 40 years and more were 2.53 times more likely to having HBV as compared to those who were 18-25 years of age. The present finding is supported by a study done in Europe [61], at the global level [68], Tanzania [58], Australia [69],and Indonesia [70]. Several explanations could account for this. One reason might be that the prevalence of Hepatitis B rises with age due to a relatively constant lifetime risk of exposure. The risk of HBV infection may have increased for older healthcare personnel due to their prolonged exposure to patients and procedures. Older workers might be more susceptible because they missed out on vaccine opportunities. Healthcare personnel may come into greater contact with blood and bodily fluids over time, increasing the risk that they will come into contact with patients who are HBV-positive. Older workers' immunity may have declined, which affects their capacity to fight off diseases like HBV. However, we can't completely rule out the possibility that, as awareness and preventative measures like donning gloves and using safety needles have grown over time, the risk of transmission has shifted [58,69].

This systematic review and meta-analysis study provided clear evidence that healthcare workers who have been diagnosed with liver disease are significantly more susceptible to hepatitis B virus infection as compared to those who are not diagnosed. This finding is consistent with previous studies done in southern Ethiopia [71], Northeast Ethiopia [72], Northwest Ethiopia [73], Northeast China [74], and Kenya [75]. The findings indicate that underlying liver conditions may compromise the immune response of these healthcare professionals, making them more vulnerable to acquiring HBV. This highlights the importance of targeted preventive measures and vaccination strategies for healthcare workers with pre-existing liver diseases, ensuring protection against this potentially serious viral infection [70]. Furthermore, a greater awareness of hepatitis B may encourage more people to get screened. For every unit of cumulative knowledge gained, researchers discovered that interest in diagnostic screening rises by 0.115 units [70,76]. The present finding is consistent with other studies done in Eastern Ethiopia [1] and Northern Gujarat [77], where training plays a key role in HBV infection reduction. According to the WHO research, by 2030, immunization, diagnostic procedures, medications, and awareness campaigns might prevent around 4.5 million preventable deaths in low- and middle-income nations. The global hepatitis plan, which aims to cut new cases of hepatitis by 90% and deaths by 65% between 2016 and 2030, has the support of every WHO member state [8,78].

Training programs enable healthcare workers better grasp the significance of adhering to safety rules by increasing awareness of HBV transmission, symptoms, and prevention techniques. Training frequently highlights how important it is to get vaccinated against HBV. Health care workers with education are more likely to be immunized and to urge their colleagues to get vaccinated. Guidelines for infection prevention, such as how to use personal protective equipment correctly, handle sharp objects safely, and lower exposure risk, are all part of effective training. Training equips health care workers with the knowledge and skills to handle exposure occurrences professionally, including prompt reporting and post-exposure treatment [77]. Continuing education ensures that HCWs are current on the latest guidelines and best practices, allowing them to adjust to new information and technologies that can reduce infection risk. Thorough training has the power to modify attitudes and behaviors about safety procedures, promoting a safety culture that places a high priority on HBV prevention [79].

The possible limitations of this study include the exclusive use of cross-sectional designs, which restricts the ability to infer causal relationships and assess temporal changes, potentially leading to biases in interpretation. High heterogeneity among the included studies may result in significant variability in effect sizes and outcomes, complicating the synthesis of findings and

limiting generalizability due to diverse populations and differing measurement methodologies. Additionally, the observed publication bias, characterized by the tendency for studies with significant or favorable results to be published more frequently, can skew the overall understanding of the research area, leading to an overestimation of effect sizes and limiting the scope of available evidence. These factors collectively hinder the robustness of the study's conclusions, underscoring the need for future research to incorporate a broader range of study designs, improve reporting standards, and address potential biases for more reliable and applicable findings.

There are important implications for practitioners, policymakers, and future researchers from this systematic review and meta-analysis. Practically speaking, a greater comprehension of HBV prevalence among HCWs can result in improved awareness and training initiatives for infection control and prevention strategies. The results may lead healthcare facilities, particularly those in high-prevalence areas, to regularly check their staff members for HBV. The review can aid in the creation of focused immunization campaigns to safeguard healthcare workers, especially those who are more vulnerable.

For those in charge of policy: Policymakers can benefit from the data by learning how much funding should go into HBV prevention and control in healthcare settings. The review's conclusions can direct the development of all-encompassing public health initiatives meant to lower the spread of HBV in healthcare settings. Lawmakers can create or improve rules and policies, such as immunization and training requirements, to guarantee better working conditions for healthcare personnel.

For future researchers: The assessment can point out areas that require more investigation, like long-term studies to determine the causal links between risk variables and HBV infection. Researchers may be motivated by the results to investigate risk factors or preventive methods that were not sufficiently explored in earlier studies.

## Conclusion

This investigation showed a significant endemicity level of HBV infection among HCWs in Africa. Age, diagnosed liver disease, and the absence of technical training were the factors significantly associated with the pooled prevalence of HBV among HCWs in Africa. Hence, providing appropriate medical follow-up and support for HCWs diagnosed with liver disease, implementing strategies to reduce the risk of HBV transmission, and providing comprehensive training and education on the epidemiology, transmission, and prevention of HBV infection play a key role in HBV infection reduction among HCWs in Africa. Early detection and diagnosis of HCWs aged 40 years and above is also an important step to preventing the risk of HBV infection among HCWs.

Moreover, educating HCWs on the importance of reporting exposures, seeking medical attention, and completing the HBV vaccination series, establishing a robust surveillance system to monitor the prevalence of HBV infection among HCWs and track the effectiveness of implemented interventions, engaging with relevant stakeholders, such as healthcare institutions, professional associations, and public health authorities, to develop and implement comprehensive policies and guidelines for HBV prevention and control among HCWs, and fostering collaboration and information-sharing among healthcare facilities to share best practices and lessons learned helps to significantly reduce the prevalence of HBV infection among HCWs in Africa setting.

## Supporting information

**S1 Table. PRISMA 2020 checklist.**
(DOCX)

**S2 Table. Results of JBI quality assessment.**
(DOCX)

**S3 Table. A numbered table of all studies.**
(DOCX)

**S4 Table. A table of all data extracted.**
(DOCX)

## Author contributions

**Conceptualization:** Leykun Berhanu, Chala Daba, Abebe Kassa Geto.

**Data curation:** Leykun Berhanu, Gete Berihun.

**Formal analysis:** Leykun Berhanu, Belay Desye.

**Funding acquisition:** Leykun Berhanu, Abebe Kassa Geto.

**Investigation:** Leykun Berhanu, Chala Daba, Gete Berihun.

**Methodology:** Leykun Berhanu, Belay Desye, Gete Berihun.

**Project administration:** Leykun Berhanu, Chala Daba, Abebe Kassa Geto.

**Resources:** Leykun Berhanu, Gete Berihun.

**Software:** Leykun Berhanu, Belay Desye, Chala Daba, Gete Berihun.

**Supervision:** Leykun Berhanu, Abebe Kassa Geto.

**Validation:** Leykun Berhanu.

**Visualization:** Leykun Berhanu, Chala Daba, Abebe Kassa Geto.

**Writing – original draft:** Leykun Berhanu, Belay Desye, Gete Berihun.

**Writing – review & editing:** Leykun Berhanu, Abebe Kassa Geto.

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
