## [Decision Letter · Decision Letter 0]

23 Oct 2024

PONE-D-24-41100Seroprevalence and Risk Factors of Hepatitis B Virus Infection among Healthcare Workers in Africa: A systematic review and Meta-analysisPLOS ONE

Dear Dr. Berhanu,

Thank you for submitting your manuscript to PLOS ONE. After careful consideration, we feel that it has merit but does not fully meet PLOS ONE’s publication criteria as it currently stands. Therefore, we invite you to submit a revised version of the manuscript that addresses the points raised during the review process.

We look forward to receiving your revised manuscript.

Kind regards,

Ashraf Elbahrawy

Academic Editor

PLOS ONE

Journal Requirements:

“The authors have no competing of interests.”

3. Please include captions for your Supporting Information files at the end of your manuscript, and update any in-text citations to match accordingly. Please see our Supporting Information guidelines for more information: http://journals.plos.org/plosone/s/supporting-information .

4. As required by our policy on Data Availability, please ensure your manuscript or supplementary information includes the following:

Reviewers' comments:

Reviewer's Responses to Questions

**Comments to the Author**

1. Is the manuscript technically sound, and do the data support the conclusions?

Reviewer #1: Yes

Reviewer #2: Partly

2. Has the statistical analysis been performed appropriately and rigorously?

Reviewer #1: Yes

Reviewer #2: Yes

3. Have the authors made all data underlying the findings in their manuscript fully available?

Reviewer #1: No

Reviewer #2: Yes

4. Is the manuscript presented in an intelligible fashion and written in standard English?

Reviewer #1: Yes

Reviewer #2: Yes

5. Review Comments to the Author

Reviewer #1: This study aimed to investigate the seroprevalence and risk factors for hepatitis B infection in healthcare workers (HCW) in Africa. The authors included 26 studies in a systematic review and meta-analysis, reporting a very high seroprevalence of infection (17.5%; 95% CI 8.36-26.04). As risk factors for infection, the study describes age >40, lack of training and previous liver disease. The authors conclude that these factors should be prioritized in interventions designed to prevent and control the occurrence of the infection.

Considering the high burden of the infection in the region, the high risk of infection in healthcare personnel, and the under-representation of African countries in studies about hepatitis B, I consider the theme extremely relevant for publication. I congratulate the authors for choosing this topic and for the effort to provide a relevant result for the African region.

The authors have properly designed the systematic review and the meta-analysis. However, the description of methodology and results should be clearer, more detailed and organized. I strongly suggest that the authors provide more details about the studies included in the review and meta-analysis, particularly regarding methodological characteristics, as a table or as supplementary material.

I also suggest that the discussion regarding risk factors for hepatitis B is improved, addressing potential biases, adding comparisons with data from HCW and from the general population, and avoiding obvious associations.

I have some comments, questions and suggestions for the authors.

ABSTRACT

1. Add the date when each source was last searched.

2. The risk of bias scale used in the study must be specified in the Abstract.

3. The total number of participants must also be specified.

4. Considering that the study evaluated seroprevalence at one moment in time, I suggest that the authors do not use the verb “develop” when referring to “having” hepatitis B. “Develop” is used several times throughout the text.

INTRODUCTION

1. Lines 63-66 should be revised, particularly regarding the proper use of the words “virus” and “infection”, which are used incorrectly (for example: “the most dangerous kind of viral hepatitis, known as hepatitis B virus”; “vaccine… to prevent this virus”).

2. Line 84: Use “donated blood”, instead of “given blood”.

3. Line 85: Use “places” (plural form).

4. Line 93: Please correct HIV as human immunodeficiency virus.

5. Line 121: Use “children”, instead of “kids”.

6. Lines 129-130: What are the differences between factors influencing HBV prevalence in previous studies?

7. Lines 146-147: As presented, this study does not assess or “highlight disparities in vaccination and treatment access”.

8. Lines 154-155: I suggest that the authors rewrite these phrases. The results of the present study can be relevant in several ways. However, saying that “the study can foster collaboration among researchers” and promote “data sharing and joint initiatives” is outside the scope of the objectives and methodology used.

9. Lines 157-158: The objectives of the study must be better stated. According to the Abstract, the study also aimed to evaluate the risk factors of hepatitis B among HCW.

MATERIALS AND METHODS

1. Further details about the search strategies should be included in the text or as supplementary files, particularly the specific search terms, dates of publication, language, and any filters used in the search.

2. Lines 175-176: The list of HCW is already detailed in lines 195-200.

3. Line 177: Use “were”, instead of “was included”.

4. Line 181: I believe the authors mean “conducted”, instead of “published”.

5. Were there any other exclusion criteria? Figure 1 mentions animal studies, language, and methodological quality.

6. Line 186: Add that 95% confidence intervals (CI) were calculated.

7. Line 192: Repeated words.

8. All data presented in the Results must be mentioned here, including all predictors that were evaluated in the study (and the specific definition, if necessary).

9. How many reviewers evaluate the data from each paper?

10. Line 218: “Perform” should be in past tense.

11. Line 220: Low risk of what?

12. Line 222: There seems to be missing words in this phrase.

13. Did all reviewers evaluate each different paper?

14. Line 229: Please correct “I2”.

15. Line 232: This line is missing a full stop.

16. Please add detailed descriptions of sensitivity analysis, subgroup analyses, and meta-regression, as well as information regarding handling of missing data.

RESULTS

1. Lines 236-237: “Result” and “study” should be in plural form.

2. Line 243: Please rewrite the part about the values reported in Nigeria; the phrase is not clear.

3. Table 1: The column “Study design” is not necessary; the information that all studies are cross-sectional can be added to the text. The 95% CI for each study must be shown. Column “Prevalence of HBV”: is it shown in percentages?

4. Lines 251-252: The choice for a random effect model should be presented and justified in the Methods section.

5. Line 274: Please mention and describe the trim and fill study in the Methods section.

6. Line 299: Why did the authors choose the sample size of 269 for the subgroup analyses?

7. Lines 331-336: The list of data extracted from the studies should be in the Methods section. Please be more specific regarding some items: diagnosed liver disease, mucocutaneous, hospitalization, dental procedure, contact history.

8. Line 342: The text is not clear.

9. Line 348: Please correct “The”.

10. Table 4: Please add the reference numbers of the studies in each line.

DISCUSSION

1. Lines 361-364: This phrase is out of context here.

2. Lines 366-368: These phrases should be rewritten. Is this study part of “evaluating the progress towards achievement of the HBV eradication target”?

3. Lines 369-374: It should be clearer that this comparison refers to studies conducted in the general population.

4. Lines 372-374: The use of parentheses must be revised. What were the 95% CIs reported in these studies? Can the authors surely state that these studies found lower prevalences of hepatitis B?

5. Line 374: Initiate a new paragraph.

6. Lines 398-402: Please provide the 95% Cis reported in the studies mentioned. I suggest that the authors carefully reflect about the comparisons made in this paragraph.

7. Lines 423-435: As stated by the authors, the seroprevalence might increase with age simply because of longer exposure to hepatitis B, regardless of the area of work. What is the seroprevalence according to age in the general population?

8. Lines 437-445: Do the “underlying liver conditions” exclude acute/chronic hepatitis B? Please make sure that hepatitis B, one of the most common “liver conditions”, are not included in these analyses.

9. Lines 449-450: I suggest that the authors rewrite this phrase and explain the “unit of cumulative knowledge gain”.

10. Line 470: There is no previous mention to missing data - please specify how missing data were handled (Methods), and specify their occurrence (Results). Regarding the limitations of the present study, please consider methodological characteristics of the included studies, heterogeneity, and publication bias.

11. Line 486: Not only medical professionals – all HCW.

12. Lines 495-498: Please be more specific. The association between the risk factors considered significant in the present study (age, liver disease, lack of training) and seropositivity for HBV is already well established.

13. How do the authors explain the wide variation of seroprevalence in different African studies, particularly regarding the extreme values, which were both obtained in Nigeria? I suggest that the authors provide a summary of methodological characteristics of each study - as a table or as supplementary material.

14. What is the seroprevalence of hepatitis B in HCW from other regions of the world?

15. Which are the risk factors for seropositivity reported in previous studies?

FIGURES 2, 9, 6, 7, 8: I suggest using smaller diamonds and expanding the horizontal axis, so that the 95% CI is more clearly visible.

FIGURE 1: I suggest changing the text direction in the purple rectangles.

Reviewer #2: Dear Authors

Thanks for conducting this a valuable systemic review and metanalysis “Seroprevalence and Risk Factors of Hepatitis B Virus Infection among Healthcare Workers in Africa: A systematic review and Meta-analysis”.

1- Introduction section

Paragraph 1 line 5

IIt is the first vaccination against hepatocellular carcinoma, a serious human cancer (2).

My comment

Is the vaccine against hepatocellular carcinoma.?

2- Methodology section

Operational definitions

Presence of HBV infection: Is the presence of serum HBsAg (current infection) and/or anti HBC (current or past resolved infection) considered evidence for exposure to HBV (recent infection or chronic carrier)

My comment

What about other markers of HBV as Anti HBs to differentiate the past resolved infection from vaccinated HCWs in those positive for anti HBC.

3- In methods for detection of HBV there are 2 modalities used ELISA and CMIA. According to your comment, there is a significant difference between 2 modalities for detection of HBV which raise the issue of missed cases in diagnosis of HBV.

4- It was mentioned that in result, he finding revealed that those who were jaundiced and diagnosed with liver disease were 5.01 times more likely to develop HBV infection as compared to those who were not exposed.

Is the liver disease related to HBV or not? If not related what was the cause of previous liver diseases.

5- Were HCWs vaccinated or not?

Regards

6. PLOS authors have the option to publish the peer review history of their article (what does this mean? ). If published, this will include your full peer review and any attached files.

**Do you want your identity to be public for this peer review?** For information about this choice, including consent withdrawal, please see our Privacy Policy .

Reviewer #1: No

Reviewer #2: **Yes: ** Ahmed Eliwa

---

## [Author Response · Author response to Decision Letter 1]

15 Jan 2025

Dear reviewers, thank you for your comment. we addressed all the comments and concerns as shown on the file attached via the submission portal.

---

## [Decision Letter · Decision Letter 1]

7 Feb 2025

PONE-D-24-41100R1Seroprevalence and risk factors of hepatitis B virus infection among healthcare workers in Africa: A systematic review and meta-analysisPLOS ONE

Dear Dr. Berhanu,

Thank you for submitting your manuscript to PLOS ONE. After careful consideration, we feel that it has merit but does not fully meet PLOS ONE’s publication criteria as it currently stands. Therefore, we invite you to submit a revised version of the manuscript that addresses the points raised during the review process.

We look forward to receiving your revised manuscript.

Kind regards,

Ashraf Elbahrawy

Academic Editor

PLOS ONE

Journal Requirements:

Reviewers' comments:

Reviewer's Responses to Questions

**Comments to the Author**

1. If the authors have adequately addressed your comments raised in a previous round of review and you feel that this manuscript is now acceptable for publication, you may indicate that here to bypass the “Comments to the Author” section, enter your conflict of interest statement in the “Confidential to Editor” section, and submit your "Accept" recommendation.

Reviewer #1: (No Response)

Reviewer #2: (No Response)

2. Is the manuscript technically sound, and do the data support the conclusions?

Reviewer #1: Yes

Reviewer #2: Yes

3. Has the statistical analysis been performed appropriately and rigorously?

Reviewer #1: Yes

Reviewer #2: Yes

4. Have the authors made all data underlying the findings in their manuscript fully available?

Reviewer #1: Yes

Reviewer #2: Yes

5. Is the manuscript presented in an intelligible fashion and written in standard English?

Reviewer #1: Yes

Reviewer #2: Yes

6. Review Comments to the Author

Reviewer #1: The authors have adequately addressed all my comments. I have only a few minor suggestions.

INTRODUCTION

Comment 1: The phrase is still not clear. I suggest saying that “the symptoms of hepatitis B virus (HBV) infection include liver inflammation (…)”.

RESULTS

Comment 6: I suggest that the criteria used to choose the sample size of 269 is added to the text.

DISCUSSION

Lines 492-494: Repeated phrase.

FIGURE 1

The edited figure was not presented in the revised manuscript.

Reviewer #2: (No Response)

7. PLOS authors have the option to publish the peer review history of their article (what does this mean? ). If published, this will include your full peer review and any attached files.

**Do you want your identity to be public for this peer review?** For information about this choice, including consent withdrawal, please see our Privacy Policy .

Reviewer #1: No

Reviewer #2: No

---

## [Author Response · Author response to Decision Letter 2]

8 Feb 2025

Response to Journal and reviewer’s comment

Journal Requirements:

Concern 1: Please review your reference list to ensure that it is complete and correct. If you have cited papers that have been retracted, please include the rationale for doing so in the manuscript text, or remove these references and replace them with relevant current references. Any changes to the reference list should be mentioned in the rebuttal letter that accompanies your revised manuscript. If you need to cite a retracted article, indicate the article’s retracted status in the References list and also include a citation and full reference for the retraction notice.

Response 1: The entire reference lists have been checked and correction has been made on volume and issue of the articles, page numbers, links to access the source, and date of access and other issues accordingly. In this study, as far our knowledge is concerned; a retracted article is not included.

Response for Reviewer’s comment

Reviewer #1: The authors have adequately addressed all my comments. I have only a few minor suggestions.

Response: Dear reviewer thank you very much for your insightful comments.

Comment 1: INTRODUCTION: The phrase is still not clear. I suggest saying that “the symptoms of hepatitis B virus (HBV) infection include liver inflammation (…)”.

Response: thank you so much. We have revised the phrase as per the given comments. The change is located on page 2 line 65.

Comment 2: I suggest that the criteria used to choose the sample size of 269 is added to the text.

Response 2: Thank you for your comment. “Two categories were used to classify the sample size: less than 269 and higher than or equal to 269. The average sample size of the included studies serves as the basis for the categorization threshold”. We believe that this idea better if included under the method section specifically on data analysis and presentation section of the revised manuscript. Therefore, the above sentences have been included under method section on page 13 line 274-276.

Comment 3: Lines 492-494: Repeated phrase.

Response 3: thank you. The repeated phrase is deleted from the revised manuscript.

Comment 4: Figure 1: The edited figure was not presented in the revised manuscript.

Response 4: We have accepted the comment. The corrected figures included in the revised manuscript.

Reviewer#2: Dear Authors Thanks for conducting this a valuable systemic review and meta-analysis “Seroprevalence and Risk Factors of Hepatitis B Virus Infection among Healthcare Workers in Africa: A systematic review and Meta-analysis”.

Response Dear reviewer, thank you so much for your comment.

Previous Comment 1: Paragraph 1 line 5. It is the first vaccination against hepatocellular carcinoma, a serious human cancer. My comment. Is the vaccine against hepatocellular carcinoma?

Previous Response 1: Dear reviewer, thank you for your comment. Normally, the vaccine is primarily against HBV. Chronic infection with HBV is a significant risk factor for developing hepatocellular carcinoma. Hence, the vaccine prevents the occurrence of HBV which in turn leads to the occurrence of hepatocellular carcinoma. It means the vaccine helps to prevent the occurrence of both HBV and its long term consequence hepatocellular carcinoma.

New Comment: please write that in the text clearly as it is mentioned in the introduction the vaccine is against hepatocellular carcinoma not against virus.

Response for new comment: Thank you for your comment. First and foremost, rather than preventing hepatocellular cancer, the vaccine described in this paragraph helps prevent HBV infection. Since hepatocellular carcinoma develops after a prolonged period of chronic HBV infection, an individual will initially be infected with the virus. Therefore, the main purpose of the vaccine to which we wish to refer in this context is to prevent HBV infection. In addition, the major concern of the study is on HBV not on its long term consequence named hepatocellular carcinoma. Therefore, we decided to remove the sentence to prevent this kind of confusion.

---

## [Editor Report · Decision Letter 2]

12 Feb 2025

Seroprevalence and risk factors of hepatitis B virus infection among healthcare workers in Africa: A systematic review and meta-analysis

PONE-D-24-41100R2

Dear Dr. Berhanu,

We’re pleased to inform you that your manuscript has been judged scientifically suitable for publication and will be formally accepted for publication once it meets all outstanding technical requirements.

Kind regards,

Ashraf Elbahrawy

Academic Editor

PLOS ONE
---

## [Editor Report · Acceptance letter]

PONE-D-24-41100R2

PLOS ONE

Dear Dr. Berhanu,

I'm pleased to inform you that your manuscript has been deemed suitable for publication in PLOS ONE. Congratulations! Your manuscript is now being handed over to our production team.

Kind regards,

on behalf of

Prof. Ashraf Elbahrawy

Academic Editor

PLOS ONE